# Assessment of the Speed and Power of Push-Ups Performed on Surfaces with Different Degrees of Instability

**DOI:** 10.3390/ijerph192113739

**Published:** 2022-10-22

**Authors:** Moisés Marquina Nieto, Jesús Rivilla-García, Alfonso de la Rubia, Jorge Lorenzo-Calvo

**Affiliations:** Facultad de Ciencias de la Actividad Física y del Deporte (INEF—Sports Department), Universidad Politécnica de Madrid, 28040 Madrid, Spain

**Keywords:** core, explosive strength, instability, resistance training, suspension training

## Abstract

(I) Training in unstable conditions, with different elements, platforms, or situations, has been used because there is a significant increase in muscle activation, balance, proprioception, and even sports performance. However, it is not known how the devices used are classified according to performance variables, nor the differences according to instability experience. (II) This study aims to analyze the differences in power and speed in push-ups with different situations of instability in trained and untrained male subjects. Power and speed in push-up exercise were analyzed in 26 untrained and 25 trained participants in 6 different situations (one stable and five unstable) (1) stable (PS), (2) monopodal (PM), (3) rings (PR), (4) TRX^®^ (PT), (5) hands-on Bosu^®^ (PH) (6) feet on Bosu^®^ (PF). The variables were analyzed using a linear position transducer. (III) The best data were evidenced with PS, followed by PR, PM, PT, PH and PF. The trained subjects obtained better results in all the conditions analyzed in mean and maximum power and speed values (*p* < 0.001). The decrease in these variables was significantly greater in the untrained subjects than in the trained subjects in the PR situation (8% and 18% respectively). In PF there were differences between groups (*p* < 0.001), reaching between 32–46% in all variables. The difference between the two groups was notable, varying between 12–58%. (IV) The results showed a negative and progressive influence of instability on power and speed in push-ups. This suggests that instability should be adapted to the subject’s experience and is not advisable in untrained subjects who wish to improve power.

## 1. Introduction

Resistance training exercises in unstable conditions, such as the use of a Swiss ball, semi-spherical balance balls and suspension devices have recently gained popularity among the athletic population. This rise in popularity stems from the fact that it significantly increases core muscle activation, balance, proprioception, and overall athletic performance [1,2,3]. Training in unstable conditions is used by trainers to increase active stabilization requirements, achieving a boost in proprioceptive activity and neuromuscular control demands. Therefore, although initially training on unstable surfaces was predominantly used in rehabilitation programmes that sought a progressive increase in proprioceptive demand, this type of training is currently included in strength and conditioning programmes [4]. In contrast, when training to increase strength or power, these characteristics should be the focus of attention and therefore strength production and speed of movement are of crucial [5,6]. The concept of “stability” could be understood as the body’s ability to maintain balance or avoid being unbalanced [7]. This stabilizing function is performed by an anatomical box consisting of several muscle groups, such as the rectus abdominis at the front, the internal and external obliques at the sides, the erector spinae, multifidus lumborum and quadratus lumborum at the back, the diaphragm at the upper edge and pelvic floor, and the psoas iliacus at the bottom called the core [8,9]. Most of the scientific literature on the importance of the core for peak performance focuses on the notion that the core is the link between the trunk and the limbs, and that an athlete is only as strong as his or her weakest link. Several studies have provided information on the importance of core training and testing in several populations [10] to reduce the risk of injury and improve performance [11,12]. In addition, they may contribute to reducing the risk of other musculoskeletal disorders in long term, which are a consequence of poor posture and sedentary lifestyles [13].

Several studies have pointed out the benefits of training in unstable conditions for enhancing performance and daily activities, referring to its specificity, because the sport is not usually practised in static conditions [1,14,15]. The use of unstable training has been proposed to improve the specific effect of movement through increased activation of the stabilizing and core muscles and is more beneficial to sports performance and daily activities [16]. Although previous research has demonstrated the benefits of resistance training in an unstable condition for trunk stability, the limb musculature’s responses to such unstable training are still a matter of discussion [17,18,19,20,21,22,23,24,25]. The analysis of the exercises executed in instability has been carried out through muscular activation [26,27,28,29,30,31,32,33] and very few studies have looked at in-depth performance variables, such as power or speed [20,24,25,34]. It has been observed that performing exercises in unstable conditions meant a significant increase in the activation of the central musculature [35] and further improvements in activation comparing unstable push-ups with standard push-ups [36]. Furthermore, using suspended push-ups caused greater activation in the main motor musculature, such as the pectoralis major, the deltoid anterior, and the brachial triceps in comparison with traditional push-ups [31], and the stabilizing musculature [37].

The concept of power has been the subject of continuous controversy due to the ambiguity of its definition and its use in sports science [38,39]. Mechanical power is often referred to as work rate [40] and is calculated by multiplying force by velocity [41]. Based on these mathematical equations, the two central components that influence an athlete’s ability to generate high power are the ability to rapidly apply higher levels of force and express high velocities of contraction. This intensity sets a limit to what is sustainable before fatigue causes the performer to slow down or reduce force application. The ability to express high power is considered one of the fundamental characteristics underlying success in various sporting activities [42,43].

Athletic movements consist of a combination of dynamic and static muscle activity. To determine whether performing strength training under unstable conditions is beneficial for athletic performance enhancement, the response of the limb musculature to unstable conditions during dynamic movement must also be examined. Considering the literature analyzed, there is no scale or progression of different unstable situations. Furthermore, there is a lack of information, evidence, and research comparing the different conditions of stability and their effect on performance variables [2]. At the same time, there is no research comparing the influence of instability according to the experience of the participants, randomly establishing its daily use without a scientific base that could support the working methods. The influence that push-up exercises would have on performance variables when compared to unstable situations has not been analyzed. Therefore, it seems necessary to investigate the acute responses, apart from muscular activation, of training under unstable conditions. The purpose of this study was to analyze the differences and changes in muscular outputs, such as power and speed during dynamic endurance exercise in different unstable conditions, compared to stable conditions in both trained and untrained male subjects. For this purpose, the dynamic flexion movement was used because it is one of the exercises most performed by athletes due to its ease of technical execution to develop strength, power, and muscular endurance of the upper limb. In addition, it is one of the main horizontal pushing exercises. The results may be useful to determine whether unstable resistance training applies to dynamic performance improvement and to establish a progression of exercises from lower to higher instability. In addition, it will demonstrate whether experience can be a determining factor in performance variables.

## 2. Materials and Methods

### 2.1. Experimental Approach to the Problem

For the design of this supervised quasi-experimental research, an ad hoc protocol was elaborated and an inter- and intra-subject comparison was used in 6 instability conditions. The situations of realization with instability were: (1) stable condition (PS; Figure 1a), (2) with single-stage execution (PM; Figure 1b), (3) with TRX^®^ (PT; Figure 1c), (4) with rings (PR; Figure 1d), (5) with feet-on Bosu^®^ (PH; Figure 1e), and (6) with hands-on Bosu^®^ (PF; Figure 1f). The equipment used in the study is the TRX Home Suspension Training Kit (Fitness Anywhere LLC, San Francisco, CA, USA) and Bosu Balance Trainer (BOSU^®^ Official Global Headquarters, Ashland, OH, USA). 

### 2.2. Participants

The sample size was calculated by analyzing statistical power (G*power 3.1.9.2, Heinrich-Heine-Universität Düsseldorf, Düsseldorf, Germany) on a mean effect size of 0.25, using a 2 × 6 repeated measures design [44,45]. Corresponding to an α-level of 0.05 and the desired power (1−β) of 0.95 at the group level, the required sample size was 28 participants. To account for the drop out throughout the study we recruited a total of 51 young adults. The 51 male participants volunteered for this study and were divided into two groups based on their previous experience with unstable training. Eligible participants had to meet the following criteria to be complied with by all subjects: (1) have a minimum of 3 years of continuous strength training experience at least 3 days a week; (2) regular use of push-up exercises in their training; (3) free of current or recent (within the last 6 months) injuries that would cause them to alter their normal physical activity; (4) abstain from any moderate to vigorous physical exercise 24 h before the experiments; (5) abstain from stimulant beverages within 12 h of study participation and any other sports supplementation. These selection criteria aimed to try to exclude the effect of external factors on the results of the experiment. All the participants are strength-trained, but the division factor between the groups was the experience in the use of instability training among the participants compared to the others. The division into groups was based on previous experience in the use of unstable devices for strength training. Table 1. shows the descriptive data of the sample.

The inclusion criteria for the group of trained athletes required a minimum of 6 months of experience with unstable devices. The time of experience of the trained subjects was based on the experience of performing strength tasks with an instability component so the tasks to improve core or the use of these materials for rehabilitation or functional recovery were not considered. This 6 months of experience was in most cases continuous with a few exceptions where there was a short break of a few weeks (1–3 weeks). There is no evidence yet to support 6 months of instability training, as the data has not yet been measured between trained and untrained subjects. The criteria are estimated based on the authors’ experience. In addition, professional, elite, or high-performance athletes were part of this group. All possible risks and benefits were explained, and written informed consent was obtained before data collection.

### 2.3. Procedures

The experimental procedures were conducted by the Declaration of Helsinki (2007) and were granted by the Ethics Committee of the Universidad Politécnica de Madrid to ensure all ethical guidelines were followed (2020-062). Besides, the research was registered on ClinicalTrials.gov (NCT04721496) (accessed on 21 January 2021).

#### 2.3.1. Experimental Protocol

The participants completed a familiarization session with different exercises to be performed as a part of the study, 48 h before the experimental protocol, supervised by the researchers. Each subject was instructed verbally, and given an explanation and demonstration of the proper technique to perform the movement of each of the different conditions.

The participants performed the exercises with self-loading, executing 2 series of 3 repetitions under 6 unstable conditions. The breaks were of 2 min after each exercise. The tasks were performed in random order. For the correct execution and control of the hand positions, three 20 kg Olympic discs were placed as a column, on which both hands had to rest. This was done so that the measurement with the encoder would be as accurate as possible and would not disturb the participants during the eccentric phase and the encoder would have a correct path during the execution. The participants started the push-ups with their arms extended (upwards) forearms and wrists in pronation, and feet at a biacromial width (shoulders). The arm was placed perpendicularly to the ground. In the prone position, the forearm and wrists were kept in pronation, while the elbow was bent at 90° and the shoulder at 45°. The hip and spine were kept in a neutral position throughout the repetitions. This movement of execution was followed for all tasks, checking that it was carried out in the correct and indicated way. The monopodal execution required participants to have their one leg raised approximately 20 cm off the ground so that it remained unsupported during the entire execution. The push-ups made with the devices in suspension were placed 20 cm from the ground with each hand holding a ring and the TRX grips. Participants were instructed to perform the tasks at the maximum possible speed. When the Bosu^®^ performance was placed on the lower body, the device was positioned with the soft, semi-spherical part facing upwards. When the instability was placed on the hands, the position was reversed, with the soft semi-spherical part facing downwards.

#### 2.3.2. Data Extraction

The variables of (1) maximum power, (2) mean power, (3) maximum speed, and (4) mean propulsive speed were analyzed in the push-up exercise, using a linear position transducer (Speed4lifts©, Madrid, Spain), used and validated by different studies [46] (Figure 2a). For the programming of the strength training load and evaluation of performance as a function of velocity, it is necessary to consider the “propulsive” phase in the concentric action. This velocity is defined as the part of the concentric phase of the movement during which the acceleration experienced by the load being moved is greater than the acceleration due to the force of −9.81 m/s^2^) [47], or in other words, the part of the movement during which the applied force is positive (>0). However, when the loads to be displaced are high (>80% 1RM), and there is no braking phase during the concentric phase, the mean velocity (MV) is an equally useful variable as the mean propulsive speed (MPS). The point is to understand that maximum, mean, and mean propulsive speed are equally reliable, but when comparing between subjects with different performances at the same absolute load (kg), the mean propulsive speed “equalizes” the potentials by disregarding the phase where no force is applied (braking phase of the load), and not averaging the value of the speed over the entire run (propulsive + braking).

The system records the changes in displacement over time, and therefore allows for the speed calculation. Power is calculated based on the speed and the changes in speed (acceleration). The linear encoder allows measuring the distance and the time to cover this distance, from which the software calculates (speed, power), in exercises of totally vertical movement, without horizontal components. To perform each push-up execution, all subjects were required to wear a waistcoat to which the encoder could be attached. The hookable end of the device was placed at the height of the sternum handle so that the device could pick up each repetition without interfering with the movement execution (Figure 2b).

#### 2.3.3. Statistical Analyses

Data analysis was performed using SPSS for Windows version 26 (IBM Corp., Armonk, NY, USA). The values shown for the quantitative and ordinal variables are the mean (M) and standard deviation (SD). The percentage (%) difference between the conditions has also been reported. Normality and homogeneity were checked through the Shapiro-Wilk test (S–W) and the Levene variance homogeneity test, respectively, and sphericity was checked through the Mauchly test. The covariance matrix was checked through the Box test. When the sphericity was not met, the Greenhouse-Geisser test was used. To check the effect of instability on the bench press, a two-factor (2 × 6) inter-subject ANOVA analysis was used. The Bonferroni test was used for multiple ex-post comparisons of the data between the different groups. To determine the differences in percentages between the participants’ data, they have been calculated taking as a reference the average data of the best performance. As an index of the effect size, η2 [48] was used. The interpretation for η2 was categorized as small for effect sizes ≥0.01 to <0.06, medium for ≥0.06 to <0.14, and large for ≥0.14 [49]. The confidence interval was set at 95% for the size effect. The significance level for all procedures was set at 0.05.

## 3. Results

### 3.1. Mean and Maximum Power

The mean and maximum power achieved by the group of trained participants was significantly higher than that recorded by the untrained ones (F1.49 = 33.51; *p* < 0.001; η2 = 0.406; F1.49 = 44.03; *p* < 0.001; η2 = 0.473, respectively). There was significant effect of the mean and the maximum power as a function of instability (F2.103 = 635.94; *p* < 0.001; η2 = 0.929; F2.111 = 604.34; *p* < 0.001; η2 = 0.925, respectively). There was effect of interaction between mean power and experience levels (F2.103 = 8.94; *p* < 0.001; η2 = 0.154; F2.111 = 11.67; *p* < 0.001; η2 = 0.192, respectively) (Table 2).

In all the conditions analyzed, the mean and maximum power reached by the trained group was significantly higher than that obtained by the untrained group (*p* < 0.001). The PS execution showed mean power and maximum power (Figure 3) values significantly higher than all other conditions (*p* < 0.001). There were differences between all situations (*p* < 0.001).

The data show notable differences between all the conditions analyzed. Thus, the stable situation is the one that obtained the best values for both mean and maximum power. In mean power, the progressive decrease was 13.03% PR, 18.65% PM, 24.18% PT, 28.38% in PH, and 38.45% in PF. At maximum power the progressive decrease was 13.30% with PR, 18.67% for PM, 23.89% with PT, 27.73% for PH, and 36.89% with PF.

Considering stable execution as a reference, the one that showed the best data, the progressive decrease of mean and maximum power for the untrained group was 17.79% and 18.85% (PR), 20.82%, and 20.94% (PM), 24.97% and 25.02% (PT), 30.29% and 30.34% (PH), and 46.19% and 40.4% (PF). For the trained group, the mean and maximum power loss was 8.68% and 8.53% (PR), 15.71% and 16.72% (PM), 21.9% and 22.91% (PT), 24.31% and 25.49% (PH), and 32.96% and 33.87% (PF).

There were significant differences between trained and untrained athletes in the values of mean and maximum power in all conditions analyzed. The differences between these two groups were as follows; mean power was 11.99% in PS, 20.77% in PR, 16.39% in PM, 13.74% in PT, 16.37% in PH, and 21.27% in PF. The differences between the two groups were that maximum power was 17.24% in PS, 26.58% in PR, 21.43% in PM, 19.50% in PT, 22.62% in PH, and 25.41% in PF.

### 3.2. Mean Propulsive and Maximum Speed

The mean propulsive speed and maximum speed reached by the trained group was significantly higher than that recorded by the untrained group (F1.49 = 540.58; *p* < 0.001; η2 = 0.917; F1.49 = 324.15; *p* < 0.001; η2 = 0.869, respectively). There was a significant effect of mean propulsive speed as a function of instability (F2.109 = 577.17; *p* < 0.001; η2 = 0.922; F2.120 = 478.20; *p* < 0.001; η2 = 0.907, respectively). An effect of the interaction between mean propulsive and maximum speed and the level of experience was observed (F2.109 = 61.07; *p* < 0.001; η2 = 0.555; F2.120 = 42.54; *p* = 0.001; η2 = 0.465 respectively) (Table 3).

In all the conditions analyzed, the mean propulsive and maximum speed reached by the trained group was significantly higher than that obtained by the untrained group (*p* < 0.001). Particularly PS execution showed significantly higher values of mean propulsive speed and maximum speed (Figure 4) than all other conditions (*p* < 0.001). There were differences between all situations (*p* < 0.001).

The data show notable differences between all the conditions analyzed. Thus, the stable situation is the one that obtained the best values for both mean propulsive and maximum speed. In mean propulsive speed, the progressive decrease was 12.20% PR, 18.29% PM, 24.39% PT, 28.05% in PH, and 36.59% in PF. At maximum speed, the progressive decrease was 12.98% with PR, 19.08% for PM, 24.43% with PT, 28.24% for PH, and 37.40% with PF.

Referring to the stable execution that showed the best data, the decrease in mean propulsive and maximum speed for the untrained group was 18.55% and 18.57% (PR), 20.23% and 20.77% (PM), 25.36% and 25.34% (PT), 29.73% and 30.36% (PH), and 41.12% and 41.09% (PF). For the trained group, the loss of mean propulsive and maximum speed was 8.93% and 9.83% (PR), 16.96% and 17.34% (PM), 23.21% and 23.70% (PT), 25.89% and 26.59% (PH), and 33.93% and 34.68% (PF).

There were significant differences between trained and untrained athletes in their mean propulsive and maximum speeds for all the conditions analyzed. The differences between the two groups were that the mean propulsive speed was 52.99% in PS, 57.96% in PR, 54.84% in PM, 54.30% in PT, 55.42% in PH, and 58.11% in PF. The differences between the two groups were that the maximum speed was 48% in PS, 53.04% in PR, 50.16% in PM, 49.11% in PT, 50.67% in PH, and 53.10% in PF.

## 4. Discussion

The purpose of this study was to analyze the differences in power and speed outcomes of push-ups under different situations of instability, both in trained and untrained male subjects. To the best of our knowledge, this is the first study that establishes a classification between different unstable situations in power and execution speed based on the experience of the athletes. The results highlighted that the situation with the best data was PS, being established PR in the second place, followed by PM, then PT, and finally the situations PH and PF. Remarkably, PR was the execution with the second-best values, ahead of PM, as it is a suspension device. However, these values may be because despite being a suspension device, the device attachment and the supports carried out are four, unlike PM, which has three, and the TRX execution, in which although there are four supports, there is only one device attachment from which the two grips then flow. In the case of rings, the attachments are independent [31].

Concerning the reduction of speed and power scores when instability was added, there is an agreement with previous studies, where they have shown that instability training does not improve the development of power or speed of movement [24,25]. Our results show a loss of power of approximately 18% for untrained athletes. Although performance variables such as power and speed have not been analyzed, even with push-up exercises, these data seem to be consistent with other studies with subjects without any previous experience of instability [50]. This decrease in power may be due to the increase in joint stiffness necessary to stabilize the joints actively involved in the exercise. Therefore, high levels of external stability [51], which is only possible when training occurs in a stable environment, such as the ground, or on a bench, seem to be necessary to improve power performance. In contrast, it has been suggested that bending with one’s body weight can be disadvantageous due to lack of resistance, our results indicate that ballistic bending without load provides sufficient external resistance to generate high movement speeds [52]. Our results have shown very high values for both mean propulsive speed (above 1.10 m/s) and maximum speed (above 1.70 m/s).

Significant speed losses concerning the stable condition have been observed for both groups. In the case of the trained subjects, this loss of speed started with a decrease of approx. 8% and reached approx. 34%. In the case of untrained athletes, it was approx. 18.5% until it reached approx. 46%. These data differ from those reported by Koshida et al. [20], where the loss of speed in the bench press with fitball was 9.1%. However, both the exercise and the device used in that study were completely different from the current study. The decrease in speed of execution, especially in the case of untrained subjects, may be due to learning new motor patterns when performing exercises in unstable conditions, which require a period of familiarization and automation of the movement [53]. It has been pointed out that any improvement in power and speed of movement in unstable conditions depends on two essential factors: the repetition of the exercise and the gradual increase of the loads [54]. This will determine the need to learn new motor patterns and adapt to improve the specificity of movements.

Another possible cause of this severe loss of power and speed, especially in untrained subjects, may be due to the factor of muscular co-activation. Generally, antagonistic muscle co-contraction increases when training on an unstable support surface, mainly to control the position of the segments when producing force [55]. Increased antagonistic activity can also occur, increasing joint stiffness, and, therefore, increasing joint stability produced by instability, thus protecting the joint complex against destabilizing external forces. Muscular co-activation of the trunk is a strategy used by the motor system to stabilize the spine [56,57]. In parallel, increased joint stiffness generated by increased antagonistic muscle co-activation when performing tasks in unstable environments may limit the production of strength, power, and speed in the extremities, because movement may be less efficient.

Another important finding is the differentiation between the classified situations, with pronounced upper and lower limb instability. In other studies, when comparing the difference between upper and lower body instability, only lower body instability produced significantly different activation values. In previous studies, there is a consistency, as the instability affects the stabilizing muscles more than the motor ones in the execution of the exercises, therefore, when placing the unstable device on the upper part of the body, it does not affect it as much as when placing the unstable device on the lower part, which challenges the stability while the movement is executed. However, the data found was with the bench press exercise [58]. Considering that bench press and push up are horizontal thrust movements that recruit similar muscle groups, the biomechanical differences between these two exercises should not be neglected. The bench press is an open-chain, single-plane exercise, whereas the effective propulsion phase of the push-up is a closed-chain, multiplanar exercise [59], so it is necessary to know the behavior of the push-up exercise.

A major consideration was that even when differences between the two groups were evident, they were more than noticeable under different conditions. In the case of power, they ranged between 11 and 26%, however, for execution speed, the differences ranged between 48 and 58%. These data could indicate that experience in this type of execution is a very decisive factor in terms of performance, both in power and above all in execution speed, where it is evident that execution control is a differential factor. Unfortunately, at the moment there is no evidence to compare the between-group values found in this study with other research analyzing unstable situations between trained and untrained people.

There were some limitations in this study. For the group of untrained subjects in the last two unstable conditions (PH and PF), a certain difficulty was observed in the execution of the exercises, although it did not prevent their complete execution. Additional evidence is needed to support this, especially as most of the available evidence indicates that when using unstable conditions improvements in strength and power are reduced compared to training in stable conditions, probably due to acute reductions in strength production, the rate of force development, leading to lower acceleration and hence lower movement velocities, and therefore lower power. In addition, it would be necessary to use a sample not only for men, but to investigates if this holds true for women. Also, it would be interesting to know its application with other exercises and movement patterns.

## 5. Conclusions

Stable execution of push-ups appears to be optimal to develop power and speed. However, the use of suspended devices with a double attachment may be of interest as an alternative to stability without excessive performance loss. The use of the Bosu^®^ has been classified as a generator of greater instability. Therefore, performance with unstable devices negatively affects speed and power output compared to tasks performed in stable conditions. Nevertheless, its affects differ among the subjects, and it seems to affect untrained athletes more. Thus, the experience in the use of different materials and unstable executions is a differential factor for performance achievement with this type of training.

## Figures and Tables

**Figure 1 ijerph-19-13739-f001:**
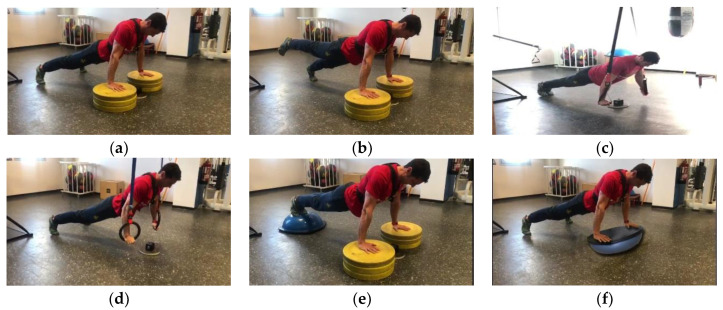
Different push-up settings during measurement: (**a**) stable; (**b**) monopodal; (**c**) suspension device: TRX^®^; (**d**) suspension device: rings; (**e**) with feet-on Bosu^®^; (**f**) with hands-on Bosu^®^.

**Figure 2 ijerph-19-13739-f002:**
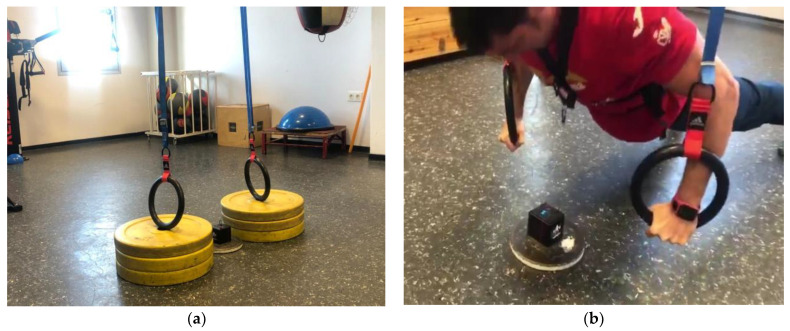
(**a**) Placement of the unstable material at the same height from the ground in both stable and unstable conditions; (**b**) Attachment of the linear velocity transducer to the waistcoat carried by the participants.

**Figure 3 ijerph-19-13739-f003:**
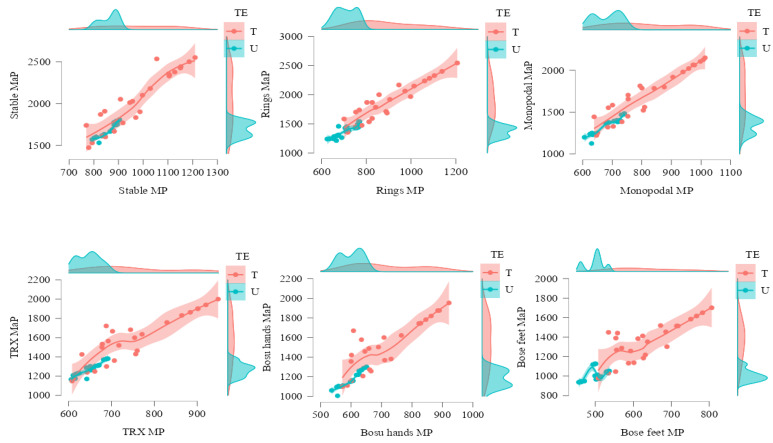
Mean power (MP) and maximum power (MaP) in the push-up exercise for untrained (U) and trained (T) groups based on experience (TE) for unstable performance tasks, measured in watts. Notes: The figure shows the two variables analyzed in each execution. On the X axis the values of mean power and on the Y axis the values of maximum power. The colors show the different groups: trained in pink and untrained in blue. The dots represent the values of the subjects and the dispersion. A line representing the trend has also been added and the colored areas are the confidence intervals. On both axes, the density of the points according to the categorization of the subjects is also represented.

**Figure 4 ijerph-19-13739-f004:**
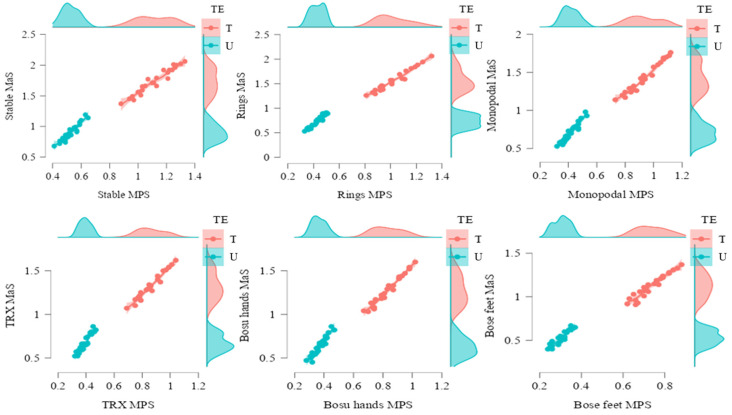
Mean propulsive speed (MPS) and maximum speed (MaS) in the push-up exercise for untrained (U) and trained (T) groups based on experience (TE) for unstable performance tasks, measured in meters/second (m/s). Notes: The figure shows the two variables analyzed in each execution. On the X axis the values of mean propulsive speed and on the Y axis the values of maximum speed. The colors show the different groups, trained in pink and untrained in blue. The dots represent the values of the subjects and the dispersion. A line representing the trend has also been added and the colored areas are the confidence intervals. On both axes the density of the points according to the categorization of the subjects is also represented.

**Table 1 ijerph-19-13739-t001:** Sample descriptive data.

Group	Age (years)	Body Mass (kg)	Body Height (cm)
Untrained (*n* = 26)	27.43 ± 5.17	77.27 ± 7.01	176.92 ± 6.01
Trained (*n* = 25)	29.16 ± 7.01	85.73 ± 17.37	183.67 ± 4.98

**Table 2 ijerph-19-13739-t002:** Descriptive statistics for mean power (MP) and maximum power (MaP) measured in watts (W) as a function of instability.

	**Untrained MP**						**Trained MP**							**Totals**		
	**N**	**M**	**SD**	**% Dif**	**IC—95%**		**N**	**M**	**SD**	**% Dif**	**IC—95%**		**% Dif Groups**	**M**	**SD**	**% Dif**
					**LL**	**UP**					**LL**	**UP**				
Stable (PS)	26	861.93	36.51		822.79	901.07	25	979.36	136.91		939.45	1019.3	11.99%	919.49	114.80	
Ring (PR)	26	708.60	45.95	17.79%	666.82	750.37	25	894.39	144.02	8.68%	851.78	936.99	20.77%	799.67	140.75	13.03%
Monopodal (PM)	26	682.46	45.10	20.82%	645.75	719.17	25	816.20	124.88	16.66%	778.76	853.64	16.39%	748.02	114.29	18.65%
TRX (PT)	26	646.72	26.75	24.97%	617.48	675.96	25	749.71	102.43	23.45%	719.89	779.52	13.74%	697.20	89.99	24.18%
Bosu Hands (PH)	26	600.85	35.97	30.29%	568.68	633.02	25	718.49	110.70	26.64%	685.68	751.29	16.37%	658.51	100.28	28.38%
Bosu Feet (PF)	26	499.74	23.81	42.02%	475.35	524.14	25	634.79	85.03	35.18%	609.92	659.67	21.27%	565.95	91.67	38.45%
	**Untrained Ma** **P**						**Trained MaP**							**Totals**		
	**N**	**M**	**SD**	**% Dif**	**IC—95%**		**N**	**M**	**DT**	**% Dif**	**IC—95%**		**% Dif Groups**	**M**	**SD**	**% Dif**
					**LL**	**UP**					**LL**	**UP**				
Stable (PS)	26	1692.96	87.37		1595.37	1790.55	25	2045.55	342.41		1946.03	2145.08	17.24%	1865.8	302.96	
Ring (PR)	26	1373.90	96.17	18.85%	1277.28	1470.51	25	1871.16	336.24	8.53%	1772.64	1969.69	26.58%	1617.65	349.17	13.30%
Monopodal (PM)	26	1338.54	100.06	20.94%	1252.20	1424.87	25	1703.55	295.88	16.72%	1615.51	1791.59	21.43%	1517.46	284.59	18.67%
TRX (PT)	26	1269.32	67.53	25.02%	1197.77	1340.87	25	1576.84	250.10	22.91%	1503.87	1649.81	19.50%	1420.07	237.51	23.89%
Bosu Hands (PH)	26	1179.35	86.23	30.34%	1104.57	1254.13	25	1524.16	256.44	25.49%	1447.90	1600.42	22.62%	1348.38	256.10	27.73%
Bosu Feet (PF)	26	1008.93	60.61	40.40%	951.51	1066.35	25	1352.68	198.77	33.87%	1294.12	1411.23	25.41%	1177.43	225.66	36.89%

Notes: MP = mean power; MaP = maximum power; M = mean; SD = standard deviation; % Dif = percentage difference between conditions; IC—95% = Interval confidence—95%; LL = lower limit; UP = upper limit; % Dif. Groups = percentage difference between groups. PS = push-up stable; PR = push-up rings; PM = push-up monopodal; PT = push-up TRX; PH = push-up hands on Bosu; PF = push-up feet on Bosu.

**Table 3 ijerph-19-13739-t003:** Descriptive statistics for mean propulsive speed (MPS) and maximum speed (MaS) measured in meters per seconds (m/s) as a function of instability.

	**Untrained MPS**						**Trained MPS**							**Totals**		
	**N**	**M**	**SD**	**% Dif**	**IC—95%**		**N**	**M**	**SD**	**% Dif**	**IC—95%**		**% Dif Groups**	**M**	**SD**	**% Dif**
					**LL**	**UP**					**LL**	**UP**				
Stable (PS)	26	0.53	0.06		0.49	0.57	25	1.12	0.12		1.08	1.16	52.99%	0.82	0.31	
Ring (PR)	26	0.43	0.05	18.55%	0.39	0.47	25	1.02	0.13	8.93%	0.98	1.06	57.96%	0.72	0.31	12.20%
Monopodal (PM)	26	0.42	0.06	20.23%	0.38	0.45	25	0.93	0.11	16.96%	0.90	0.97	54.84%	0.67	0.27	18.29%
TRX (PT)	26	0.39	0.04	25.36%	0.37	0.42	25	0.86	0.09	23.21%	0.83	0.89	54.30%	0.62	0.25	24.39%
Bosu Hands (PH)	26	0.37	0.05	29.73%	0.34	0.40	25	0.83	0.10	25.89%	0.80	0.86	55.42%	0.59	0.25	28.05%
Bosu Feet (PF)	26	0.31	0.04	41.12%	0.28	0.33	25	0.74	0.07	33.93%	0.72	0.76	58.11%	0.52	0.23	36.59%
	**Untrained MaS**						**Trained MaS**							**Totals**		
	**N**	**M**	**SD**	**% Dif**	**IC—95%**		**N**	**M**	**DT**	**% Dif**	**IC—95%**		**% Dif Groups**	**M**	**SD**	**% Dif**
					**LL**	**UP**					**LL**	**UP**				
Stable (PS)	26	0.90	0.14		0.83	0.97	25	1.73	0.21		1.67	1.81	48.00%	1.31	0.46	
Ring (PR)	26	0.73	0.12	18.57%	0.66	0.80	25	1.56	0.22	9.83%	1.49	1.63	53.04%	1.14	0.45	12.98%
Monopodal (PM)	26	0.71	0.12	20.77%	0.65	0.78	25	1.43	0.19	17.34%	1.36	1.49	50.16%	1.06	0.39	19.08%
TRX (PT)	26	0.67	0.10	25.34%	0.62	0.72	25	1.32	0.15	23.70%	1.27	1.38	49.11%	0.99	0.35	24.43%
Bosu Hands (PH)	26	0.63	0.11	30.36%	0.57	0.68	25	1.27	0.17	26.59%	1.22	1.33	50.67%	0.94	0.35	28.24%
Bosu Feet (PF)	26	0.53	0.08	41.09%	0,49	0.58	25	1.13	0.13	34.68%	1.08	1.17	53.10%	0.82	0.32	37.40%

Notes: MPS = mean propulsive speed; MaS = maximum speed; M = mean; SD = standard deviation; % Dif = percentage difference between conditions; IC—95% = Interval confidence—95%; LL = lower limit; UP = upper limit; % Dif. Groups = percentage difference between groups. PS = push-up stable; PR = push-up rings; PM = push-up monopodal; PT = push-up TRX; PH = push-up hands on Bosu; PF = push-up feet on Bosu.

## Data Availability

Not applicable.

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
