# Peer review of "Assessment of the Speed and Power of Push-Ups Performed on Surfaces with Different Degrees of Instability"

_ijerph, 2022, doi:10.3390/ijerph192113739_

Round 1

Reviewer 1 Report

An interesting study that enjoyed reading. Practically focused and it was easy to see how the study translates to the real world. I see this as a strength of the study. However, the point that trained people are stronger than untrained in somewhat expected, and I know that stability is a way to create the interesting question, but all too frequently, the obvious results have an overpowering effect. My suggestion is to re-write the results and get the main effect of trained v untrained out of the way promptly and then focus on the effects of stability. This is important as the message is getting lost. 

Minor points

a) Instability is clumsily put in the abstract but explained much better in the introduction. Please re-visit.

b) Untrained - why 6 months? And many people have stop-start relationship with exercise. What about the motivation of participants. Please justify 6 months. 

c) The criterion for inclusion is detailed. How did they check people truly met it; not drank coffee for 12 hours? was this really checked and how? Self-report is acceptable but if you relied on it, then its a limitation. 

Reviewer 2 Report

Dear Authors, all comments are in the appendix. 

Round 2

Reviewer 1 Report

I enjoyed re-reading this revised paper. The authors considered my comments and have revised the paper accordingly. 

One point worth making is that they should acknowledge that the 6-month of exercise set as a criterion is an arbitrary time period. They have estimated what they think is a sufficient period of time. They should acknowledge this decision - it does not change the quality of the paper, but the is not strong evidence behind the choice of 6-months. It is an estimated period of time. 

Author Response

Thank you very much for your appreciation. We recognise that there is still no evidence regarding the 6 months of instability training as the differences between trained and untrained in this area have not yet been observed. This criterion has been estimated based on the experience of the authors. It will be reflected in the paper from line 142 onwards, as follows:

"There is no evidence yet to support 6 months of instability training, as the data has not yet been measured between trained and untrained subjects. The criteria is estimated based on the authors' experience."

Reviewer 2 Report

Dear Authors, the paper is almost acceptable, but I still need some completions.
